# Modeling and Optimization of Ultrasound-Assisted Extraction of Bioactive Compounds from *Allium sativum* Leaves Using Response Surface Methodology and Artificial Neural Network Coupled with Genetic Algorithm

**DOI:** 10.3390/foods12091925

**Published:** 2023-05-08

**Authors:** Shubhra Shekhar, Prem Prakash, Poonam Singha, Kamlesh Prasad, Sushil Kumar Singh

**Affiliations:** 1Department of Food Process Engineering, National Institute of Technology, Rourkela 769008, Odisha, India; shekharshubhra@gmail.com (S.S.); poonamsingha2652@gmail.com (P.S.); 2Department of Food Engineering and Technology, Sant Longowal Institute of Engineering and Technology, Longowal 148106, Punjab, India; pprakashft@gmail.com

**Keywords:** *Allium sativum*, optimization, ANN-GA, response surface methodology, ultrasound-assisted extraction, polyphenols, flavanoids, antioxidant

## Abstract

This study explains the effect of ultrasound on the extraction of the bioactive compounds from garlic (*Allium sativum* L.) leaf powder. The experiment was carried out by varying the ultrasound amplitude (30–60%), treatment time (5–15 min), and ethanol concentration (40–60%) required to obtain the maximum extraction yield of total phenol content (TPC), total flavonoid content (TFC), and antioxidant activity. Rotatable central composite design (RCCD) provided experimental parameter combinations in the ultrasound-assisted extraction (UAE) of garlic leaf powder. The values of extraction yield, TPC, TFC, and antioxidant activity for the optimized condition of RSM were obtained at 53% amplitude, 13 min of treatment time, and 50% ethanol concentration. The values of the target compounds predicted at this optimized condition from RSM were 32.2% extraction yield, 9.9 mg GAE/g TPC, 6.8 mg QE/g TFC, and 58% antioxidant activity. The ANN-GA optimized condition for the leaf extracts was obtained at 60% amplitude, 13 min treatment time, and 53% ethanol concentration. The predicted values of optimized condition obtained by ANN-GA were recorded as 32.1738% extraction yield and 9.8661 mg GAE/g, 6.8398 mg QE/g, and 58.5527% for TPC, TFC, and antioxidant activity, respectively. The matured leaves of garlic, if not harvested during its cultivation, often go waste despite being rich in antioxidants and phenolic compounds. With the increased demand for the production of value-added products, the extraction of the bioactive compounds from garlic leaves can resolve waste management and potential health issues without affecting the crop yield through the process for high-end use in value addition.

## 1. Introduction

The agro-food sector plays a strategic role in the commercial and economic operations of producing and cultivating *Allium* species. *Allium* is one of the most produced and consumed vegetative products worldwide, with a world production of 32.4 MMT in the year 2020, with China being the highest producer, followed by India [1].

Recently, the demand for natural compounds in the prevention and cure of various diseases such as asthma, cholera, fever, diabetes, hypertension, and cancer is constantly increasing [2,3]. Besides that, consumers are becoming more health conscious and more aware of the ingredients in food products and are becoming inclined towards natural food additives rather than synthetic ones. This shift is causing an increase in the demand for natural compounds in the industrial processing of food*. Allium* species is not only a popular vegetable but also has therapeutic values [4]. It has been used in traditional medicinal practices since ancient times, attributed mainly to sulfur-containing compounds [5,6]. *Allium sativum* L. belongs to the onion family Alliaceae, having volatile compounds, sulfur, antioxidants, and many bioactive compounds [7]. With the plant growth, the older matured leaves, if not harvested, are dried on the plant and go to waste. Harvesting those leaves helps in the protection of plants from lodging and provides economic importance if harvesting occurs at proper intervals during their cultivation. Leaves of *A. sativum* contain higher levels of bioactive compounds, i.e., polyphenols, ascorbic acid, and dietary fiber, and exhibit more antioxidant activity [8]. Similar results were observed for the aged leaf extracts from garlic for antioxidant activity, followed by the flower and then for the bulb [9]. The presence of antioxidants in food can prevent cell damage by scavenging free radicals, thus preventing various ailments [3,4,10]. Furthermore, more than 20 phenolic compounds are present in garlic, which is a higher number than that in most vegetables [11]. β-resorcylic acid is the major phenolic compound in garlic, accompanied by pyrogallol, gallic acid, rutin, protocatechuic acid, and quercetin [12,13].

The bioactive compounds mainly find application in the development of newer food products, medicine, agriculture, and industry but also are used in the green synthesis of metal nanoparticles [14,15,16,17,18,19]. In the green synthesis of metal nanoparticles, a bioactive compound can serve as a reducing agent by donating electrons and converting metal ions into metal nanoparticles (MNPs) and as a stabilizing agent by preventing the agglomeration of the MNPs [14,15,19]. The bioactive compounds in the extract form are mainly used to synthesize the MNPs [20]. The bioactive compounds present in garlic can be derived in the form of an extract by using different conventional or novel extraction methods. The conventional extraction techniques used have lower efficiency than novel techniques of extraction such as microwave-assisted extraction (MAE) and ultrasound-assisted extraction (UAE) [21,22,23,24]. Unlike high-power ultrasound applied in component extraction, low-power ultrasound is often used as a non-destructive tool in food component analysis and medical imaging [25,26,27]. The extraction using UAE is relatively more suitable than other methods due to its cost-effectiveness and lower instrumental requirements [28]. High-power ultrasound is used to achieve increased extraction efficiency due to the phenomena of cavitation bubbles. These cavitation bubbles compress and decompress with ultrasonic waves and finally collapse, producing a shock wave. These shock waves can enhance the mixing and mechanical effect, causing increased solvent diffusion in the product matrix and subsequently increasing the release of intracellular compounds to the bulk medium [29,30]. To obtain the maximum yield of the intracellular compound and superior extraction, the optimum values of the factors influencing the extract must be determined [31]. Prediction tools such as RSM and ANN-GN are primarily utilized to determine the optimal conditions because of their ability to model complex relationships between variables and predict outcomes based on those relationships [32].

Response surface methodology (RSM) is a mathematical and statistical method used to elucidate the relationship between multiple input variables (also known as factors) and a single output variable (also known as the response). It is often used in engineering and scientific research to optimize the design of a product or process to identify the optimal combination of input variables that will result in the desired output [33,34,35]. The mathematical model that explains the input variables and the response variable is used to make predictions and to identify the optimal combinations of input variables [36,37]. In contrast, an artificial neural network (ANN) is a machine learning method inspired by the structure and function of the human brain. The structure comprises neurons that receive, process, and transmit information [37]. Based on the input data, an ANN is designed to recognize patterns and make predictions. It can be used as a prediction tool even when the data are complex and have a non-linear relationship which requires high levels of prediction accuracy [38,39]. To evaluate the optimized condition from the predicted parameters, a genetic algorithm (GA) is applied.

A GA is an optimization technique that is built on the natural selection and genetics principle. It comprises the application of genetic operators such as selection, crossover, and mutation to create a new generation of candidate solutions [40]. The main aim of a genetic algorithm is to find the optimum solution to the problem by mimicking the natural selection process [41]. An ANN combined with a GA is a hybrid approach to machine learning that combines the capabilities of both the ANN and GA. Combining both allows the incorporation of a combined approach and strengthens prediction and optimization. For the analysis and processing of data, a neural network can be used, whereas for optimizing the parameters, a genetic algorithm can be used [42].

Thus, the aim of this study was to assess and compare the effectiveness of two popular optimization techniques, RSM and ANN-GA, in the modeling and optimization of complex systems for the production of quality garlic (*A. sativum*) leaf powder extract. Moreover, the effect of different extraction parameters on the properties of the extract was also investigated.

## 2. Materials and Methods

### 2.1. Reagents and Standards

To estimate the bioactive compounds in the sample, chemicals such as quercetin dihydrate, Folin–Ciocalteu reagent, gallic acid, DPPH, and other chemicals were procured from Hi-Media Laboratories Pvt. Ltd., Mumbai, India. All the glassware used was from Borosil Glass Works Limited, Mumbai, India.

### 2.2. Raw Material

The two most matured green leaves from each *A. sativum* plant leaving four well-developed leaves on the plant of variety Haryana Garlic-17 (HG-17) were harvested in the early morning from a locally grown progressive farmer’s field near the vicinity of SLIET. Longowal Campus, Sangrur (Punjab, India). Before use, the hay-like dried portion at the distal ends of harvested green leaves was removed and then washed in running tap water twice. After apparent surface water drying, the leaves were cut lengthwise into approximately 1.5 cm and dried using a PID-controlled cabinet drier maintained at 50 ± 1 °C. The dried garlic leaves were ground and sieved using a sieve of 100 mesh to obtain garlic leaf powder (Figure 1).

### 2.3. Ultrasound-Assisted Extraction (UAE)

UAE was carried out using a 20 kHz probe ultrasonicator (Qsonica 700, Qsonica, Newtown, CT, USA) with a probe diameter of 16 mm. A frequency of 20 kHz with a constant power of 700 W was used for the extraction of the target compounds. The temperature of the samples was maintained by keeping them in an ice bath, causing the temperature not to exceed 40 °C. For the experimental runs, the garlic leaf powder was mixed with an ethanol–water mixture and placed in a beaker, maintaining the same distance from the probe for each run. The applied energy for each treatment was recorded as the instrument’s energy reading (J). Calorimetric energy was calculated as per Kikuchi and Uchida [43] for each experiment. The digital panel on the instrument was used to control the ultrasound amplitude and treatment time. After applying the ultrasound, the extract obtained was immediately filtered using the Whatman filter paper 42. The obtained extracts were kept at 4 °C for further analysis.

### 2.4. One Factor at a Time

The one factor at a time (OFAT) approach was used to identify and select critical factors among the various independent variables affecting the extraction yield. The various experimental variables and their levels selected via OFAT experiments were ultrasonic amplitude (20%, 30%, 40%, 50%, and 60%), ethanol concentration (20%, 40%, 50%, 60%, 80%, and 100%), treatment time (5, 10, 15, and 20 min) and solvent-to-solid ratio (10:1, 20:1, 30:1, 40:1, and 50:1). During this approach only a single factor was varied, keeping other variables constant. The total phenol content of different experiments was evaluated for the selection of independent variables that significantly influenced the extraction efficacy.

### 2.5. Determination of Proximate Composition

Moisture, protein, fat, ash content, and pH of the garlic leaves were estimated using the standard method [44]. The carbohydrate content of garlic leaves was determined using the difference method.

### 2.6. Determination of Total Phenol Content

For determining the total phenol content, the Folin–Ciocalteu method was followed [45] with slight modification. Initially, 0.1 mL aliquot of the extract was combined with 2.5 mL Folin–Ciocalteu reagent. The mixture was then incubated for 30 min after adding 2 mL of sodium carbonate (10% *w*/*v*). The absorbance of the standard and the reaction mixture was observed using a spectrophotometer (UV-1800 Shimadzu spectrophotometer, Shimadzu, Kyoto, Japan) at 725 nm.

### 2.7. Determination of Total Flavonoid Content

Total flavonoid content was determined using the aluminum chloride colorimetry assay method [46]. As the standard in this procedure, quercetin was utilized. Initially, 0.5 mL of supernatant was mixed with 4 mL of distilled water. Then, 0.3 mL of 5% sodium nitrite was added to the above mixture. After 5 min, a mixture of 0.3 mL of 10% aluminum chloride was added. At the 6th minute, 2 mL of 1 M sodium hydroxide was added, and 2.4 mL of distilled water was mixed properly. The mixture was kept at room temperature for 15 min, and then absorbance was measured at 510 nm using a spectrophotometer. The calibration curve of quercetin was drawn from 0–1 mg/mL, and the results were expressed as quercetin equivalent.

### 2.8. DPPH Assay

The antioxidant activity was evaluated using a modified version of the DPPH scavenging assay as proposed in [47]. For this, 0.1 mL of garlic leaf extract was mixed with 3.9 mL of 1 mM DPPH solution. The mixture was maintained in the dark at room temperature for thirty minutes. The absorbance of the solution was then evaluated with a spectrophotometer at 515 nm. For the preparation of the control, instead of the sample, 0.1 mL of distilled water was combined with DPPH solution (3.9 mL) and observed at the same wavelength. Equation (1) was used to calculate the DPPH scavenging activity.
(1)$$\mathrm{DPPH}\;\mathrm{scavenging}\;\left(\%\right)=\frac{\left(A_{S}-A_{C}\right)}{A_{C}}\times 100$$
where, *A_C_* and *A_S_* are the absorbances of the control and extract, respectively.

### 2.9. Estimation of Organosulfur and Phenolic Compounds Using HPLC

Organosulfur and phenolic compounds were estimated by using a chromatographic estimation technique. It was performed using Water’s HPLC 2489 equipped with Water’s UV detector (Waters Corporation, Milford, MA, USA). The Luna LC column (150 × 4.6 mm, 5 μ particle size) and C18 column (Phenomix 100 × 4.6 mm, 5 μ particle size) were used for the estimation of organosulfur compounds (OSCs) and phenolic compounds, respectively.

OSCs were detected with a gradient condition of 50% mobile phase A (acetonitrile) and 50% mobile phase B (water) with a constant flow rate of 1.0 mL/min at a detection wavelength of 254 nm. Phenolic compounds were detected at 280 nm wavelength with varying concentrations of solvent A (water/acetonitrile/formic acid = 96/3.9/1; *v*/*v*) and solvent B (acetonitrile/formic acid = 99.9/0.1; *v*/*v*) with time, at the flow rate of 1.0 mL/min. The gradient program was performed with varying percentage proportions of solvent A and solvent B (at the initial stage, the concentration of solvent A:solvent B was 100:0, at 5 min 95:5, 25 min 85:15, 30 min 80:20, 39 min 75:25, 43 min 55:45, 48 min 5:95, 50 min 5:95, 55 min 80:20, and 60 min 100:0).

All the prepared solvents were ultrasonicated for 7–10 min to remove the entrapped air before the analysis. The extracted samples were filtered through a 0.45 μ nylon filter to eliminate the undesirable contaminants if present. An individual OSC and phenolic standard was first run to estimate the faithful retention time at respective wavelengths. These compounds were identified by comparison with standard compound retention times. The individual OSCs and phenolic compounds were calculated by applying Equations (2) and (3), respectively.
(2)$$\mathrm{Concentration}\;\mathrm{of}\;\mathrm{individual}\;\mathrm{OSC}\;\left(ppm\right)=\frac{Area_{sample}}{Area_{Std}}\times Conc_{std}$$
(3)$$\mathrm{Concentration}\;\mathrm{of}\;\mathrm{individual}\;\mathrm{phenolic}\;\mathrm{compound}\;\left(ppm\right)=\frac{Area_{sample}}{Area_{Std}}\times Conc_{std}$$

### 2.10. Experimental Design

The input parameters and their ranges were designated based on the method of one factor at a time. The solute-to-solvent ratio of 1:40 was kept constant throughout the experiment. Amplitude (20–60%), time (5–15 min), and ethanol concentration (40–60%) were chosen as independent variables to optimize the extraction yield, total phenol content, total flavonoid content, and antioxidant activity as the dependent parameters. Rotatable central composite design (RCCD) was used to obtain the different compositions of input variables and analyzed using design expert software (version 11.0.5.0, Stat-ease Inc., Minneapolis, MN, USA). According to RCCD, a total of 20 runs consisted of experimental combinations of eight factorial, six axial, and six central points [48] as presented in Table 1 and Table 2.

### 2.11. Fourier Transform Infrared Spectroscopy Analysis

Fourier transform infrared spectroscopy (FTIR) is an analytical technique that provides information on the functional groups present in the extract, which helps to identify and quantify the chemical components present in the sample. A Bruker Alpha FTIR system (Germany) was used to determine the chemical conformation of the garlic leaf powder extract obtained using UAE. The samples were scanned over a range of 4000–500 cm^−1^ at a resolution of 4 cm^−1^ with 64 scans per sample.

### 2.12. Field Emission Scanning Electron Microscopy (FE-SEM)

Field emission scanning electron microscopy (FE-SEM) is a powerful imaging technique used to visualize the pores or microcracks present on the surface structure of the sample at higher resolution. The structural analysis of the garlic leaf powder with and without ultrasonic treatment was observed using a field emission scanning electron microscope (FEI Novanano SEM 450, Nebraska Center for Materials and Nanoscience, Lincoln, NE, USA). A magnification of 2500× was used to acquire the garlic leaf powder image before and after the application of ultrasound treatment.

### 2.13. Response Surface Methodology (RSM)

RSM was utilized to model and optimize the process parameters to optimize the dependent parameter values. The response variables were fitted to the second-order polynomial model as given in Equation (4). The experimental data were analyzed using a second-order full polynomial regression model to predict optimal conditions.
(4)$$Y=\beta_{0}+\overset{k}{\underset{i=1}{\sum }}\beta_{i}X_{i}+\overset{k}{\underset{i=1}{\sum }}\beta_{ii}X_{i}^{2}+\overset{k-1}{\underset{i}{\sum }}\overset{k}{\underset{j}{\sum }}\beta_{ij}X_{i}X_{j}$$
where *Y* represents the response variable; *X_i_* and *X_j_* are actual values in coded form of the independent variable; *k* represents the number of considered factors (*k* = 3); and $\beta_{0}$_,_ $\beta_{i}$, $\beta_{jj}$, and $\beta_{ij}$ are the constant and coefficients of linear, quadratic, and interaction terms, respectively.

Analysis of variance (ANOVA) was used to determine the significance of the regression coefficient, and the developed regression models were validated using the conducted statistical analysis. The adequacy of the model was evaluated by the coefficient of multiple determination (*R*^2^). The terms in the polynomial were considered statistically different when *p* < 0.05.

### 2.14. Artificial Neural Network (ANN) Modeling Coupled with Genetic Algorithm (GA)

The artificial neural network and the genetic algorithm were used for modeling and optimization of parameters by neural net fitting application in machine learning and the optimization tool in maths, statistics, and optimization tool, respectively, of MATLAB software (Version R2018a, The MathWorks, Inc., Natick, MA, USA). The neural network used for the experiment comprised one input, one hidden, and one output layer. Three neurons were present in the input layer, i.e., ultrasound amplitude (*X*_1_), treatment time (*X*_2_), and ethanol concentration (*X*_3_). A single hidden layer was selected for the experiment to prevent the issue of overfitting. The number of neurons in the hidden layer was decided based on the trial and error method, i.e., the heuristic method [49] to obtain the larger *R*^2^ and lower MSE values; based on that, 10 neurons were selected for the hidden layer. The training for each response (TPC (*Y*_1_), TFC (*Y*_2_), Antioxidant (*Y*_3_), and Extraction yield (*Y*_4_)) was performed separately, and hence the output layer comprised only a single neuron. RCCD was used to build the experimental design for UAE of garlic leaf powder with the above three input parameters. The Levenberg–Marquardt backpropagation algorithm was used for training the algorithm as it is fast and most recommended in the toolbox. The input–output datasets were divided into three subsets, namely training, validation, and testing, consisting of 70%, 15%, and 15%, respectively. The transfer function for the hidden layer was the hyperbolic sigmoid function (*tansig*), whereas the transfer function for the output layer was linear (*purelin*). The network was trained continuously until the maximum *R*^2^ value and minimum MSE value were attained. After training, bias and the weight values of the generated model were obtained, and Equation (5) was utilized for the prediction of the response values (*Yi*).
(5)$$tansig\left(n\right)=\frac{2}{1+e^{-2\times n}}-1$$
(6)$$purelin\left(n\right)=n$$
(7)$$Y_{i}=purelin(W_{OH}\times tansig(U_{IH}\times X_{i}+TH)+TO)$$
where *n* is the net input, *Y_i_* is the predicted output parameter, and *X_i_* is the input parameter. *U_IH_* is the weight between the input layer and hidden layers, and *W_OH_* is the weight between the hidden layer and output layer. *TH* is the bias value of hidden layer neurons, and *TO* is the bias value of output.

The GA was used to optimize the input variables of the developed ANN model. During the optimization phase, the fitness function (*f*) was employed to maximize all of the dependent parameters. The migration, crossover function, mutation function, selection function, scaling function, creation function, and population type for the genetic algorithm were chosen as forward migration, scattered, adaptive feasibility, roulette function, rank, feasible population, and double vector. For the best optimization result, a 0.8 value was used for the crossover fraction, and the remaining functions were set to default. The ANN-GA optimization fitness function (*f*) is as follows:(8)$$f=-(Y_{1}+Y_{2}+Y_{3}+Y_{4})$$
where *Y*_1_, *Y*_2_, *Y*_3_, and *Y*_4_ are the ANN predicted extraction yield, TPC, TFC, and antioxidant response, respectively, and the negative sign indicates maximizing of the function *f* in the GA.

### 2.15. Statistical Analysis

All experiments were performed at least in triplicates; one-way analysis of variance (ANOVA) was used to examine the data, and SPSS (version 25) statistic software was used for the analysis. Duncan’s multiple range test was used to separate mean values, and significant differences were considered at *p* < 0.05. Data are presented as mean ± standard deviation (SD). The results were analyzed using Design Expert (version 11.0.5.0, Stat-ease Inc.), and ANOVA was used to determine the model’s significant terms and regression coefficients of linear, interaction, and quadratic terms.

Statistical parameters such as coefficient of multiple determination (*R*^2^), normal root mean square (NRMSE), root mean square error (RSME), mean percentage error (MPE), normal mean square error (NMSE), mean square error (MSE), and average absolute deviation (AAD) were used for the comparison of the performance of models built with ANN and RSM. The optimum model for expressing the responses has the highest *R*^2^ and lowest NRMSE, RSME, MPE, AAD, and MSE. The formulae of *R*^2^, NRMSE, RSME, MPE, AAD, and MSE are presented as follows:(9)$$\mathrm{MSE}=\frac{\sum {\left(x_{p}-x_{a}\right)}^{2}}{n}$$
(10)$$\mathrm{NMSE}=\frac{\mathrm{MSE}}{x_{m}}$$
(11)$$\mathrm{AAD}=\frac{\sum \left|x_{p}-x_{a}\right|}{n}$$
(12)$$\mathrm{MPE}=\frac{100}{n}\sum |\frac{\left(x_{p}-x_{a}\right)}{x_{p}}|$$
(13)$$\mathrm{RMSE}=\sqrt{\frac{\sum {\left(x_{p}-x_{a}\right)}^{2}}{n}}$$
(14)$$\mathrm{NRMSE}=\frac{\mathrm{RMSE}}{x_{m}}$$
(15)$$R^{2}=1-\frac{\sum {\left(x_{p}-x_{a}\right)}^{2}}{\sum {\left(x_{p}-x_{m}\right)}^{2}}$$
where *x_p_*, *x_a_*, *x_m_*, and *n* represent predicted data, experimental data, experimental mean data, and number of experiments, respectively.

## 3. Results and Discussion

### 3.1. Proximate Analysis

The chemical parameters of fresh garlic leaves (variety: HG-17) were analyzed for the proximate composition and pH. The moisture content of garlic was found to be 90.04%, which is in the range of other green leafy vegetables [50]. Protein, fat, carbohydrate, ash content, and pH were found to be 2.23%, 0.18%, 5.99%, 1.56%, and 6.08%, respectively (Table 3). Protein content in the garlic leaves was found to be considerably higher and fat content was lower than that in other leafy vegetables, namely *Cardiospermum halicacabum*, *Premna latifolia*, *Argyreia pomacea*, *Mollugo pentaphylla*, and *Pisonia grandis* [51].

### 3.2. One Factor at a Time (OFAT)

The range for the independent variable was determined using the OFAT approach. The range of the independent variable was selected in such a way that it covers the expected operating range and is not wide enough to make the experimental design impractical and complex. Preliminary screening of the independent variable was performed firstly by keeping all the variables constant and varying only the solute-to-solvent ratio. The solid-to-solvent ratio was kept constant at 1:40 as there was no significant effect on the extraction efficiency. After keeping the solid-to-solvent ratio constant, other independent variables were adjusted in such a way as to include the optimal values for each factor, i.e., ultrasound amplitude, treatment time, and ethanol concentration. These identified optimal values of selected critical variables were considered as the respective levels of the central point in the CCD of RSM (Table 1). The optimal values identified in the OFAT approach for ultrasonic amplitude, treatment time, and ethanol concentration were 45%, 10 min, and 50%, respectively.

### 3.3. RSM Analysis

#### 3.3.1. Model Fitting

The experimental and predicted values of all the dependent variables (extraction yield, TPC, TFC, and antioxidant content) and the independent parameters (ultrasound amplitude, time, and ethanol concentration) along with the applied energy and calorimetric energy are presented in Table 2. Similarly, the equations below represent the second-order full polynomial models with regression coefficients used to predict the dependent variables.
EY = 31.91 + 0.4514*X*_1_ + 0.5161*X*_2_ + 0.3259*X*_3_ − 0.0950*X*_1_*X*_2_ − 0.2875*X*_1_*X*_3_ − 0.5175*X*_2_*X*_3_ − 0.3482*X*_1_^2^ − 0.3482*X*_2_^2^ − 0.3058*X*_3_^2^(16)
TPC = 9.85 + 0.1194*X*_1_ + 0.1052*X*_2_ + 0.0735*X*_3_ + 0.0163*X*_1_*X*_2_ + 0.0187*X*_1_*X*_3_ − 0.0662*X*_2_*X*_3_ − 0.0794*X*_1_^2^ − 0.0511*X*_2_^2^ − 0.0740*X*_3_^2^(17)
TFC = 6.71 + 0.4526*X*_1_ + 0.2183*X*_2_ + 0.2596*X*_3_ − 0.0188*X*_1_*X*_2_ + 0.0337*X*_1_*X*_3_ − 0.0188*X*_2_*X*_3_ − 0.1614*X*_1_^2^ − 0.1985*X*_2_^2^ − 0.2038*X*_3_^2^(18)
Antioxidant activity = 57.84 + 1.01*X*_1_ + 0.6629*X*_2_ + 1.35*X*_3_ + 0.1412*X*_1_*X*_2_ − 0.3113*X*_1_*X*_3_ − 0.3438*X*_2_*X*_3_ − 0.6993*X*_1_^2^ − 0.6481*X*_2_^2^ − 0.6092*X*_3_^2^(19)
where *X*_1_, *X*_2_, and *X*_3_ are independent variables in coded form for ultrasound amplitude, treatment time, and ethanol concentration, respectively.

The regression coefficients, *R*^2^, adj. *R*^2^, and lack of fit of the obtained second-order polynomial equation for the various responses of garlic leaf extract treated with ultrasound are shown in Table 4. Table 5 shows different parameters such as *R*^2^, NRMSE, RSME, MPE, AAD, and MSE using formulae given in Equations (7)–(13).

According to Table 4, the *R*^2^ and adj. *R*^2^ values of EY, TPC, TFC, and antioxidant activity were 0.96 and 0.93, 0.96 and 0.92, 0.93 and 0.87, and 0.91 and 0.84, respectively, indicating that the generated model is desirable. In addition, the lower values of statistical parameters for the responses (Table 4) verify the model’s desirability. All four responses exhibited a non-significant lack of fit, with values of 0.660 (EY), 0.336 (TPC), 0.051 (TFC), and 0.109 (antioxidant), indicating that the models fit the data well.

#### 3.3.2. Effect of Ultrasound Treatment on Total Phenol Content

Table 2 shows the mean response value of total phenol content (TPC) of garlic leaf extract for 20 different experimental runs. The highest TPC value, 9.9 mg GAE/g, was obtained at an amplitude of 45%, treatment time of 10 min, and ethanol concentration of 50%, whereas the lowest value, 9.38 mg GAE/g, was obtained at an amplitude of 30%, treatment time of 5 min and ethanol concentration of 40%. Ultrasound amplitude (*X*_1_) (*p* < 0.001), treatment time (*X*_2_) (*p* < 0.001), and ethanol concentration (*X*_3_) (*p* < 0.001) exhibited a linear significant positive effect towards extraction of TPC. Similarly, the interaction term *X*_2_*X*_3_ (*p* < 0.05) and each quadratic term substantially impacted the TPC yield.

The three-dimensional surface plots shown in Figure 2a–c describe the interaction between amplitude (*X*_1_) and treatment time (*X*_2_). TPC yield increases as amplitude and treatment time increase; this may be due to increased cavitation and thermal effect at higher ultrasonic amplitude, which induces disruption and softening of the cell walls [52]. However, in the case of interaction between ethanol concentration (*X*_3_) and treatment time (*X*_2_), total phenol content yield initially increased with an increase in ethanol concentration but subsequently decreased; this could be due to the disintegration of the cell wall as water acts as a swelling agent breaking up the plant cell wall. The ethanol dissolves the polyphenol by disrupting the solvent solid contact area, but a further increase in the ethanol concentration leads to an increase in the dielectric constant of the solvent, which causes a further decrease in the yield of TPC [53]. Shirzad et al. [54] reported a similar trend for extracting TPC from olive leaves using ultrasound.

#### 3.3.3. Effect of Ultrasound Treatment on Total Flavonoid Content (TFC)

The highest TFC value of 6.96 QE/g was obtained at an amplitude of 60%, treatment time of 15 min, and ethanol concentration of 60%, whereas the lowest value, 5.39 QE/g, was obtained at an amplitude of 30%, treatment time of 5 min, and ethanol concentration of 40%. Antioxidant activity was linearly, significantly, and positively affected by ultrasound amplitude (*X*_1_) (*p* < 0.005), treatment time (*X*_2_) (*p* < 0.001), and ethanol concentration (*X*_3_) (*p* < 0.001) and negatively affected by the quadratic terms.

The three-dimensional surface plot represented in Figure 2 shows the interaction between amplitude (*X*_1_) and time (*X*_2_), which shows an increase in the value of TFC with the increase in the applied amplitude and time (up to about 12 min). The increase in the phenolic compounds can be due to the cavitation effect caused by US disruption of the structure of leaves, which aided in the diffusion of phenolic compounds in the solvent [55]. Further increase in the time caused a reduction in the TFC value. A similar trend was reported by Zimare et al. [56], where ultrasound treatment beyond 10 min decreased the TFC content of *Lobelia nicotianifolia* leaf extracts. The reduction in the bioactive compounds with treatment time can be attributed to the formation of free radicals during cavitation [57].

#### 3.3.4. Effect of Ultrasound Treatment on Antioxidant Activity

The DPPH (2,2-diphenyl-1-picrylhydrazyl) scavenging assay is a commonly used method for measuring the antioxidant activity of a substance. The mean values of the obtained experimental data and ANOVA analysis are shown in Table 2 and Table 4, respectively. The highest antioxidant value obtained was 58.59% at amplitude, treatment time, and ethanol concentration of 45%, 10 min, and 50%, respectively, whereas the lowest value obtained was 53.12% at amplitude, treatment time, and ethanol concentration of 45%, 10 min, and 33%, respectively. Antioxidant activity was positively affected by ultrasound amplitude (*X*_1_) (*p* < 0.001), treatment time (*X*_2_) (*p* < 0.01), and ethanol concentration (*X*_3_) (*p* < 0.001) and negatively affected by the quadratic terms.

The three-dimensional plot in Figure 2 represents the interaction between amplitude (*X*_1_) and time (*X*_2_), demonstrating an increase in antioxidant activity with an increase in amplitude and time. The antioxidant values increased up to a certain ultrasonic amplitude of 55%. The surge in the antioxidant activity might be due to cavitation that increases the thermal effect leading to disruption of the structure of plant cells that leads to the release of previously bound antioxidants within the cell [52]. Moreover, increased antioxidant activity can also be attributed to increased polyphenol content in garlic leaf extract due to cavitation generated during ultrasonication [58]. The further decrease in antioxidant activity is due to the detrimental effect of oxidation during prolonged ultrasonication [59]. A similar trend for antioxidant activity in cashew apple bagasse was reported [31].

#### 3.3.5. Effect of Ultrasound Treatment on Extraction Yield

The mean value of extraction yield is presented in Table 2. The highest value of 32.24% was obtained at amplitude, treatment time, and ethanol concentration of 60%, 15 min, and 40%, respectively, whereas the lowest extraction yield of 28.71% was obtained at an amplitude of 30%, treatment time of 5 min, and ethanol concentration of 40%. The ANOVA analysis of the data represents a positive effect of linear terms, i.e., ultrasound amplitude (*X*_1_) (*p* < 0.0001), treatment time (*X*_2_) (*p* < 0.0001), and ethanol concentration (*X*_3_) (*p* < 0.0005), and negative effect on interaction terms such as the amplitude–ethanol concentration (*X*_1_*X*_3_) (*p* < 0.01) and treatment time–ethanol concentration (*X*_2_*X*_3_) (*p* < 0.0001) and quadratic terms.

Visual analysis of the three-dimensional surface and contour plots show how different parameters affect the extraction yield. The plot showing the interaction between amplitude (*X*_1_) and treatment time (*X*_2_) with ethanol concentration (*X*_3_) shows that there is an increase in the extraction yield with an increase in ethanol concentration to 53%, but further decrease can be observed due to the negative effect of the quadratic term on the ethanol concentration. The decrease in the yield can be due to the fact that at higher ethanol concentrations, dehydration of plant tissue occurs, which results in a decrease in the yield at higher concentrations [60]. A similar trend was reported by Tomšik, Pavlić, Vladić, Ramić, Brindza, and Vidović [29] for bioactive compound extraction using *Allium ursinum* L.

### 3.4. Artificial Neural Network Analysis

#### Model Fitting

The Levenberg–Marquardt (LM) algorithm was used for the prediction of TPC, TFC, antioxidant, and extraction yield. Maximum *R*^2^ and minimum MSE values give reliability and applicability to the developed model in prediction adequacy. Accordingly, the best model was selected based on the maximum *R*^2^ and minimum MSE values. The final model that was selected for TPC, TFC, and antioxidant activity was obtained at epoch 5, and for the extraction yield, epoch 6. The model had one layer of input (3 neurons), one hidden layer (10 neurons), and one layer of output for each response. The neural network diagram for the experiment is depicted in Figure 3.

The values of MSE for training, testing, and validation of TPC were 0.53554, 0.45408, and 0.38106, and the values of *R*^2^ for the same were 0.984, 0.987, and 0.988, respectively. Similarly, for TFC, the values of MSE for training, testing, and validation were 0.33468, 0.66318, and 0.931, and the values of *R*^2^ was 0.987, 0.994, and 0.994, respectively. Further, the values of MSE for training, testing, and validation of antioxidant activity were 0.62286, 0.8568, and 0.13710, and the values of *R*^2^ for the same were 0.962, 0.996, and 0.884, respectively. Similarly, the MSE values for training, testing, and validation of extraction yield were 0.18343, 0.77744, and 0.12273, and the values of *R*^2^ for the same were 0.967, 0.993, and 0.987, respectively. High *R*^2^ values confirm the prediction capability of the model. The optimized output data obtained from the model were compared with the target values, and the difference was used to determine the weights for the minimization of the individual sum of square errors (SSE). Figure 4 shows the error histogram and performance of the model.

### 3.5. Optimization Using Genetic Algorithm (GA)

Using GA, the process parameters (ultrasound amplitude, treatment time, and ethanol concentration) were tuned to maximize the extraction of TPC, TFC, antioxidant activity, and extraction yield. The procedure was repeated until the minimum RSME and MSE between the individual fitness values and mean value were achieved. After the mutation, the optimization cycle was maintained, and if the targeted outcome was not reached, the entire population was employed for reproduction, crossover, and mutation for the following cycle. Figure 5 illustrates the relation between the number of generations and the fitness value.

After the 63rd generation, the figure did not show any variation in the fitness value; a value of −107.413 was obtained as the mean fitness value. The optimization was accomplished under the following conditions: treatment period, 13 min; amplitude, 60%; and ethanol concentration, 53%, with applied energy of 26,150 J and calorimetric energy of 161.37 J. The expected results of EY, TPC, TFC, and antioxidant activity were obtained at this optimum condition as 32.17%, 9.87 mg GAE/g, 6.84 mg QE/g, and 58.55%, respectively.

### 3.6. Comparison between RSM and ANN Models

The predicted values of RSM and ANN models were statistically compared using statistical measures, such as *R*^2^, NRMSE, RSME, MPE, AAD, and MSE.

Statistical parameters of both models are shown in Table 5, from which it can be inferred that both models were entirely predictive. However, when compared, the ANN model’s overall *R*^2^ values were more significant than those of RSM. In addition, the ANN model has lower values for NRMSE, RSME, MPE, AAD, and MSE than the RSM model. These findings indicate that the artificial intelligence approach of the ANN model predicts responses far more accurately than the RSM model, the statistical approach in optimization.

The values of extraction yield, TPC, TFC, and antioxidant activity for the optimized condition of RSM were obtained at 53% amplitude, 13 min of treatment time, and 50% ethanol concentration with 22,430 J as applied energy and 163.94 J as calorimetric energy. The values of the target compounds predicted at this optimized condition were 32.2% extraction yield, 9.9 mg GAE/g TPC, 6.8 mg QE/g TFC, and 58% antioxidant activity. Whereas for the ANN-GA optimized condition of 60% amplitude, 13 min treatment time, and ethanol concentration of 53%, having applied energy of 26,150 J and calorimetric energy of 161.37 J, the expected values of EY, TPC, TFC, and antioxidant activity were 32.1738%, 9.8661 mg GAE/g, 6.8398 mg QE/g, and 58.5527%, respectively. The experimental values at these conditions showed that the values obtained at the optimum condition determined by ANN-GA were more significant than those obtained using RSM as ANN predicted more accurately as it had a higher *R*^2^ value and lower NRMSE, RSME, MPE, AAD, and MSE values than RSM.

### 3.7. FE-SEM Analysis

Figure 6 shows the image of the garlic leaf powder at 2500× magnification before and after the application of ultrasound treatment. It was observed that there was a significant change in the garlic leaf powder surface after the ultrasonic treatment application because of the formation of microcracks, pores, and cell wall fragmentation. These changes were caused due to cavitation on the surface-treated leaf extract samples and are depicted in Figure 6b. These microcracks, pores, and cell wall fragmentation assist the extraction of phenolic compounds present in the plant cell’s innermost part by increasing the material’s surface area and mass transfer of the target compounds to the solvent. Similar results were reported by AL-Bukhaiti et al. [61], showing that the application of ultrasound treatment on *Cissus rotundifolia* increased the phenol diffusion rate from the plant cell into the solvent. Therefore, the benefits of applying ultrasound may result from tissue and cell damage that would speed up the mass transfer of bioactive molecules such as polyphenols.

### 3.8. FTIR, Polyphenol, and Sulfur Compound Analysis

The infrared spectra of garlic leaf extract obtained at the optimum condition of UAE revealed the presence of functional groups that indicate a polyphenol-rich extract. The spectral region between 3570 and 3200 cm^−1^ represents stretching vibrations of the hydroxyl group (OH) (Figure 7) [62]. The bands between 3300 and 2500 cm^−1^ confirmed the presence of alkane (C-H stretch) and carboxylic acid (O-H stretch) [63]. The region of 1711–1511 cm^−1^ shows C=O stretching [64]. The peak in the region between 1600 and 1500 cm^−1^ could be due to aromatic nuclei vibration, and that between 1500 and 1400 cm^−1^ indicates the presence of C–C cm^−1^ bonds in the phenolic groups [65]. The spectral region between 1044 cm^−1^ and 1088 cm^−1^ shows the presence of an alcohol functional group [66].

### 3.9. Identification and Quantification of Organosulfur Compounds (OSCs)

Table 6 shows the concentrations of different identified organosulfur compounds as observed in the chromatogram of garlic leaf powder extract, which were confirmed as alliin, S-allyl-L-cysteine (SAC), and allicin. The concentrations of alliin, SAC, and allicin were estimated to be 90.207, 4.314, and 219.536 ppm, respectively. Allicin is the major bioactive compound formed on crushing of garlic and is decomposed at the injection port of GC and converted to vinyldithiins [67]; thus, an HPLC detection approach for the organosulfur compounds is preferred.

### 3.10. Identification and Quantification of Phenolic Compounds

Twelve phenolic compounds were identified in the HPLC chromatogram against the used known phenolic compound standard (Figure 8). The retention times of standard phenolic compounds were 4.225 min, 7.333 min, 13.532 min, 14.201 min, 16.479 min, 23.587 min, 26.421 min, 26.772 min, 30.292 min, 31.905 min, 42.509 min, and 50.179 min against the known concentrations of standard compounds, namely gallic acid, 3,4-dihydroxybenzoic acid, chlorogenic acid, catechin hydrate, syringic acid, p-coumaric acid, rutin, ellagic acid, benzoic acid, hesperidin, quercetin, and β-carotene, respectively. Table 6 shows the concentration of identified phenolic compounds present in the extracted sample. Among the identified phenolic compounds, β-carotene (117.607 ppm) showed the maximum concentration of identified phenolic compounds. Chlorogenic acid, 3,4-dihydroxybenzoic acid, catechin hydrate, rutin, benzoic acid, and gallic acid were found to have concentrations of 36.537 ppm, 34.403 ppm, 26.327 ppm, 21.741 ppm, 15.234 ppm, and 13.591 ppm, respectively. The result obtained showed a similar trend for the phenolic compounds for gallic acid, catechin hydrate, and p-coumaric acid and showed lower concentrations for the phenolic compounds syringic acid and benzoic acid, which were estimated in the aqueous extracts of garlic cultivars [68]. Syringic acid, p-coumaric acid, ellagic acid, hesperidin, and quercetin were found in concentrations of 4.276 ppm, 1.228 ppm, 0.968 ppm, 9.272 ppm, and 6.140 ppm, respectively.

## 4. Conclusions

Ultrasound-assisted garlic leaf powder bioactive component extraction was optimized using response surface methodology and an artificial neural network coupled with genetic algorithm (ANN–GA) approach. Analysis of variance data for assessing the impact of different independent parameters on the EY, TPC, TFC, and antioxidant content were analyzed, and the developed models were assessed for predicting the responses in accordance with the experimental results. The artificial neural network coupled with genetic algorithm approach was much more efficient in prediction and estimation capabilities even with fewer datasets as compared to RSM, making it a more reliable tool. The optimum values of the independent variables obtained using ANN-GA were 60% ultrasound amplitude, 13 min treatment time, and 53% ethanol concentration, having applied energy of 26,150 J and calorimetric energy of 161.37 J. The expected values for this optimized condition were obtained as 32.1738%, 9.8661 mg GAE/g, 6.8398 mg QE/g, and 58.5527% for EY, TPC, TFC, and antioxidant activity, respectively. The results also showed the potential of garlic leaf powder in the extraction of free radical scavenging characteristics. FTIR and FE-SEM further revealed the positive effect of applied high-power ultrasound in releasing active components, specifically polyphenols and sulfur-containing compounds. This paves the way for the high-end applications of developed garlic leaf powder and extract. Scaling up the developed process thus may have the potential to unlock a multitude of benefits in the exploration of bioactive compounds derived from garlic leaves, preferably at the industrial level.

## Figures and Tables

**Figure 1 foods-12-01925-f001:**
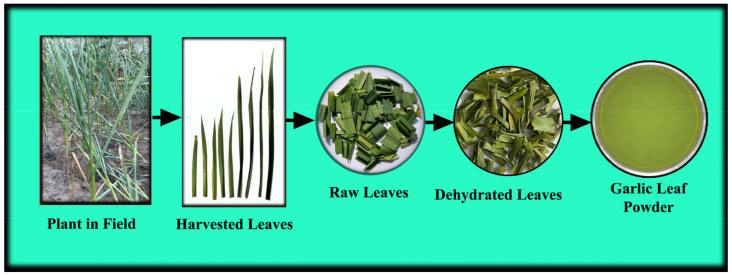
Process flow diagram for garlic leaf powder preparation.

**Figure 2 foods-12-01925-f002:**
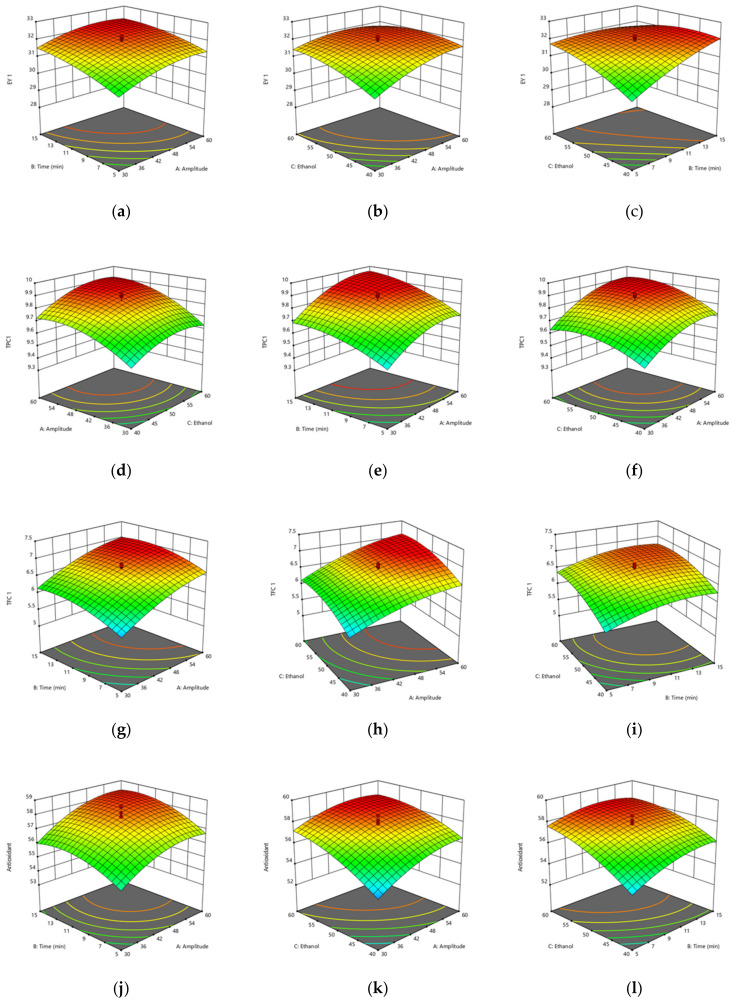
The effect of ultrasound amplitude, extraction time, and ethanol concentration on (**a**–**c**) extraction yield, (**d**–**f**) total phenol content, (**g**–**i**) total flavonoid content, and (**j**–**l**) antioxidant activity of garlic leaf powder.

**Figure 3 foods-12-01925-f003:**
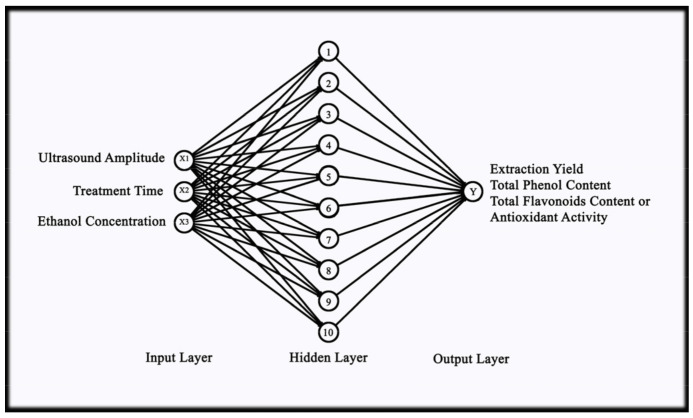
Optimal neural network architecture for present experiment.

**Figure 4 foods-12-01925-f004:**
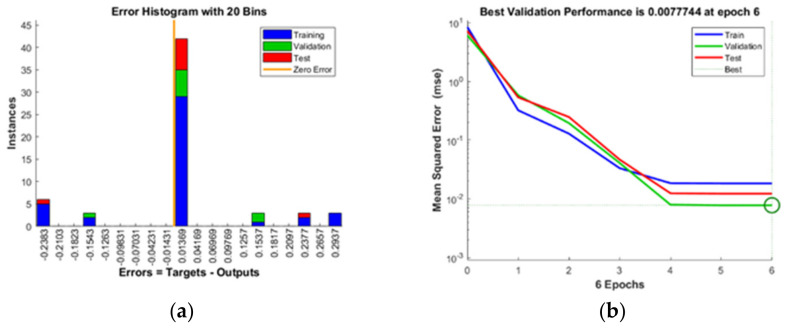
Post-training performance and error histogram of (**a**,**b**) extraction yield, (**c**,**d**) total phenol content, (**e**,**f**) total flavonoid content, and (**g**,**h**) antioxidant activity of generated artificial neural network model.

**Figure 5 foods-12-01925-f005:**
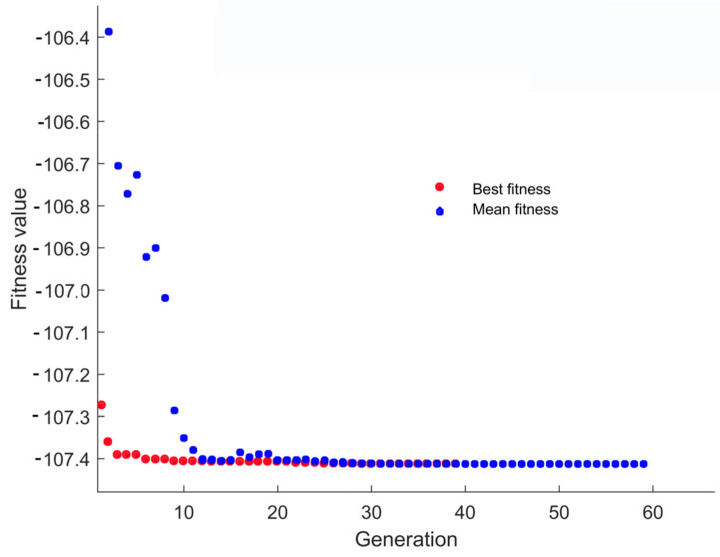
Fitness value variation across generations during optimization with a genetic algorithm.

**Figure 6 foods-12-01925-f006:**
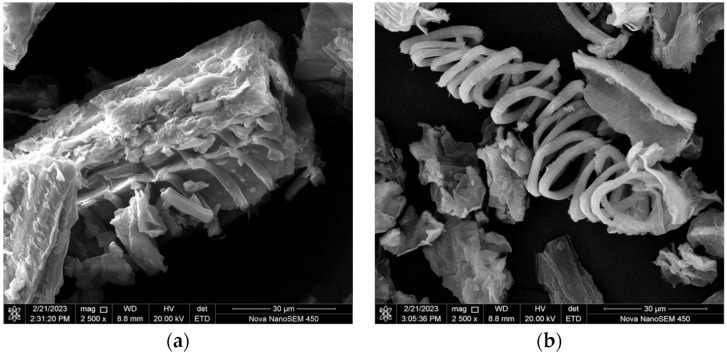
FE-SEM images of garlic leaf powder (**a**) before treatment and (**b**) after treatment by ultrasonic processing at optimized ANN-GA condition (60% ultrasound amplitude, 13 min treatment time, and 53% ethanol concentration).

**Figure 7 foods-12-01925-f007:**
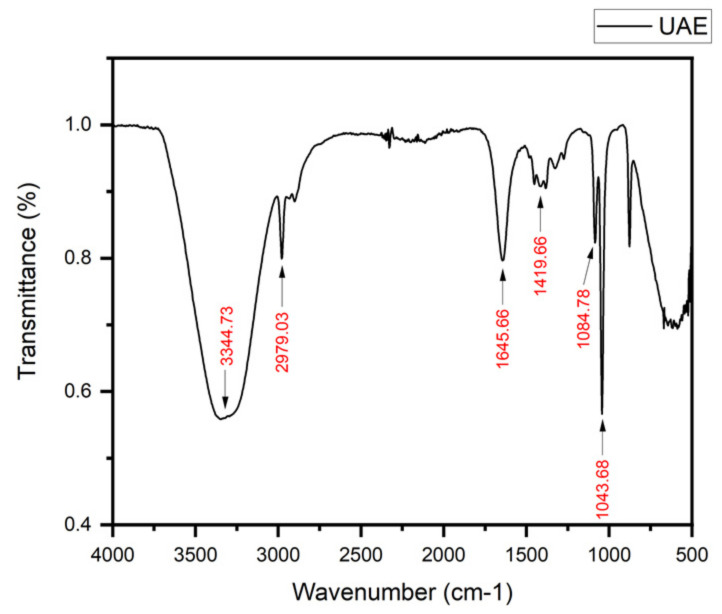
FTIR analysis of the UAE extract at optimized ANN−GA condition (60% ultrasound amplitude, 13 min treatment time, and 53% ethanol concentration).

**Figure 8 foods-12-01925-f008:**
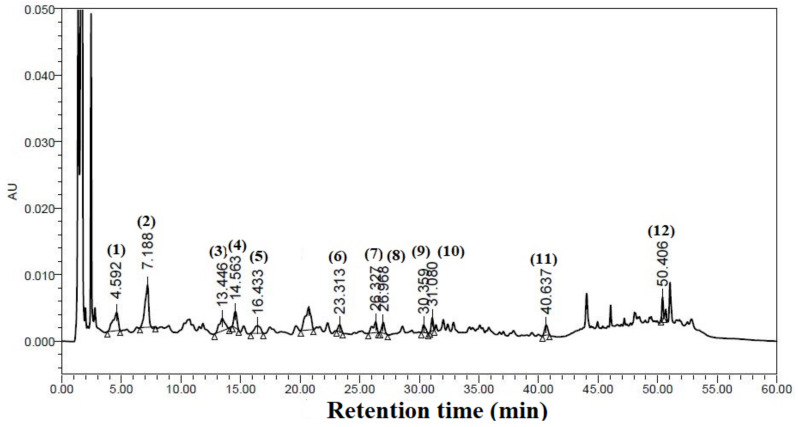
HPLC chromatogram of identified polyphenols of garlic leaf powder.

**Table 1 foods-12-01925-t001:** The experimental domain of rotatable central composite design.

		Coded Level				
Independent Variable	Units	−α*	−1	0	1	α*
Ultrasound Amplitude (*X*_1_)	%	19.77 (~20)	30	45	60	70.23 (~70)
Treatment Time (*X*_2_)	min	1.60 (~2)	5	10	15	18.40 (~18)
Ethanol Conc. (*X*_3_)	%	33.18 (~33)	40	50	60	66.82 (~67)

* The (±) α values of the independent parameters are represented as round figures.

**Table 2 foods-12-01925-t002:** Rotatable central composite design of three factors with their experimental and predicted responses using RSM and ANN.

Run	Space Type	Ultrasound Amplitude (X1)	Treatment Time (X2)	Ethanol Conc. (X3)	Applied Energy (J) *	Calorimetric Energy (J) **	Yield (%)			TPC (mg GAE/g)			TFC (mg QE/g)			Antioxidant (%)		
							Experimental	RSM Pred.	ANN Pred.	Experimental	RSM Pred.	ANN Pred.	Experimental	RSM Pred.	ANN Pred.	Experimental	RSM Pred.	ANN Pred.
1	Center	45	10	50	16,744	294.34	31.65	31.91	31.90	9.81	9.85	9.85	6.80	6.71	6.69	57.04	57.84	57.85
2	Center	45	10	50	16,744	277.73	32.14	31.91	31.90	9.89	9.85	9.85	6.78	6.71	6.69	57.74	57.84	57.85
3	Factorial	60	5	40	11,187	302.98	30.65	30.51	30.65	9.49	9.48	9.49	6.32	6.09	6.32	55.23	54.70	55.23
4	Center	45	10	50	16,744	837.19	31.74	31.91	31.90	9.82	9.85	9.85	6.85	6.71	6.69	57.87	57.84	57.85
5	Factorial	30	15	60	18,366	97.68	31.16	31.29	31.16	9.63	9.60	9.63	5.97	6.14	5.97	56.55	56.71	56.55
6	Factorial	30	5	60	6261	188.38	31.26	31.11	31.26	9.57	9.56	9.57	5.72	5.70	5.72	56.93	56.35	56.93
7	Factorial	30	5	40	7390	173.16	28.71	28.84	28.71	9.38	9.31	9.38	5.39	5.21	5.39	53.21	52.34	53.21
8	Axial	45	2	50	2846	166.50	30.45	30.42	30.45	9.47	9.53	9.47	5.49	5.78	5.49	53.86	54.89	53.86
9	Axial	70	10	50	23,389	336.38	31.73	31.68	31.73	9.82	9.82	9.82	6.88	7.01	6.88	57.80	57.55	57.80
10	Axial	20	10	50	10,390	134.66	30.11	30.17	30.11	9.38	9.42	9.38	5.53	5.49	5.53	53.39	54.16	53.39
11	Factorial	30	15	40	22,332	219.78	31.31	31.10	31.31	9.62	9.63	9.62	5.68	5.72	5.68	54.39	54.07	54.39
12	Axial	45	10	67	13,877	158.85	31.62	31.59	31.62	9.75	9.76	9.75	6.69	6.57	6.69	58.58	58.38	58.58
13	Axial	45	18	50	33,414	116.80	32.12	32.16	32.12	9.89	9.88	9.89	6.71	6.51	6.71	57.62	57.12	57.62
14	Center	45	10	50	16,744	170.09	32.21	31.91	31.90	9.90	9.85	9.85	6.67	6.71	6.69	58.59	57.84	57.85
15	Center	45	10	50	16,744	173.16	32.05	31.91	31.90	9.88	9.85	9.85	6.54	6.71	6.69	57.73	57.84	57.85
16	Factorial	60	15	40	30,957	109.41	32.24	32.39	32.24	9.88	9.86	9.88	6.57	6.52	6.57	56.79	56.99	56.79
17	Axial	45	10	33	17,820	166.50	30.46	30.50	30.46	9.48	9.52	9.48	5.48	5.70	5.48	53.12	53.84	53.12
18	Factorial	60	5	60	10,050	173.16	31.42	31.62	31.42	9.84	9.80	9.84	6.82	6.71	6.82	57.52	57.47	57.52
19	Factorial	60	15	60	28,857	146.52	31.57	31.43	31.57	9.88	9.91	9.88	6.96	7.07	6.96	57.89	58.39	57.89
20	Center	45	10	50	16,744	173.16	31.67	31.91	31.90	9.80	9.85	9.85	6.63	6.71	6.69	58.13	57.84	57.85

Abbreviations: EY, extraction yield; TPC, total phenol content; TFC, total flavonoid content; Antioxidant, antioxidant activity; RSM pred., response surface methodology predicted value; ANN, artificial neural network predicted value; * applied energy obtained from instrument; ** calorimetric energy calculated as per Kikuchi and Uchida [43].

**Table 3 foods-12-01925-t003:** Proximate composition of raw garlic leaves.

Chemical Components	Amount (%)
Moisture	90.04
Protein	2.23
Fat	0.18
Ash	1.56
Carbohydrate	5.99

**Table 4 foods-12-01925-t004:** Estimated coefficients of the fitted model and analysis of variance table representing the regression coefficients, *R*^2^, adj. *R*^2^, and lack of fit of the responses.

Coefficient	Yield (%)	TPC (mg GAE/g)	TFC (mg QE/g)	Antioxidant (%)
b_0_	+31.91	+9.85	+6.71	+57.84
b_1_	+0.4514 *	+0.1194 *	+0.4526 *	+1.01 *
b_2_	+0.5161 *	+0.1052 *	+0.2183 *	+0.6629 *
b_3_	+0.3259 *	+0.0735 *	+0.2596 *	+1.35 *
b_12_	−0.0950	+0.0163	−0.0188	+0.1412
b_13_	−0.2875 *	+0.0187	+0.0337	−0.3113
b_23_	−0.5175 *	−0.0662 *	−0.0188	−0.3438
b_11_	−0.3482 *	−0.0794 *	−0.1614 *	−0.6993 *
b_22_	−0.2192 *	−0.0511 *	−0.1985 *	−0.6481 *
b_33_	−0.3058 *	−0.0740 *	−0.2038 *	−0.6092 *
*R* ^2^	0.963	0.962	0.932	0.917
Adj. *R*^2^	0.930	0.928	0.870	0.842
Lack of fit	0.660	0.336	0.051	0.109

* Significant at *p* ≤ 0.05. Abbreviations: Yield, extraction yield; TPC, total phenol content; TFC, total flavonoid content. b denotes the regression model; b_0_ denotes the constant term; b_1_, b_2_, and b_3_ denote the linear terms; b_12_, b_13_, and b_23_ denote the interaction terms; and b_11_, b_22_, and b_33_ denote the quadratic terms. 1, 2, and 3 signify independent parameters, such as ultrasound amplitude, treatment time, and ethanol concentration.

**Table 5 foods-12-01925-t005:** List of all statistical parameters (*R*^2^, NRMSE, RSME, MPE, AAD, and MSE) of responses.

	EY		TPC		TFC		Antioxidant	
	RSM	ANN	RSM	ANN	RSM	ANN	RSM	ANN
RMSE	0.1630	0.1259	0.0353	0.0224	0.1442	0.0604	0.5298	0.2558
NMSE	0.0052	0.0040	0.0036	0.0023	0.0228	0.0095	0.0094	0.0045
MSE	0.0266	0.0158	0.0012	0.0005	0.0208	0.0036	0.2807	0.0655
NRMSE	0.0008	0.0005	0.0001	0.0001	0.0033	0.0006	0.0050	0.0012
MPE	0.4500	0.2099	0.3160	0.1218	2.0121	0.4388	0.7860	0.1799
AAD	0.1416	0.0670	0.0306	0.0120	0.1248	0.0295	0.4376	0.1040
*R* ^2^	0.9633	0.9781	0.9622	0.9848	0.9317	0.9883	0.9171	0.9807

Abbreviations: EY, extraction yield; TPC, total phenol content; TFC, total flavonoid content, Antioxidant, antioxidant activity; RSM, response surface methodology; ANN, artificial neural network; *R*^2^, coefficient of multiple determination; NRMSE, normal root mean square; RSME, root mean square error; MPE, mean percentage error; NMSE, normal mean square error; MSE, mean square error; AAD, average absolute deviation.

**Table 6 foods-12-01925-t006:** Concentration of organosulfur and phenolic compounds in garlic leaf powder using HPLC at 254 nm and 280 nm.

Peak Number Compounds			Concentration (ppm)
Organosulfur		Alliin	90.207
		S-Allyl-L-cysteine	4.314
		Allicin	219.536
Phenolic	1	Gallic acid	13.591
	2	3,4-Dihydroxybenzoic acid	34.403
	3	Chlorogenic acid	36.537
	4	Catechin hydrate	26.327
	5	Syringic acid	4.276
	6	p-Coumaric acid	1.228
	7	Rutin	21.741
	8	Ellagic acid	0.968
	9	Benzoic acid	15.234
	10	Hesperidin	9.272
	11	Quercetin	6.140
	12	β-Carotene	117.607

## Data Availability

All research data has been reported in this manuscript.

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
