# Peer review of "Modeling and Optimization of Ultrasound-Assisted Extraction of Bioactive Compounds from Allium sativum Leaves Using Response Surface Methodology and Artificial Neural Network Coupled with Genetic Algorithm"

_foods, 2023, doi:10.3390/foods12091925_

Round 1

Reviewer 1 Report

1.    The authors changed the three parameters in rather wide ranges: the amplitude of ultrasound  (30-60%), treatment time  (5-15  min), and ethanol  concentration  (40-60%). But according to Table 2, the results changed very little in response to these changes of conditions: phenols by ±3% (9.4 – 9.9 mg\g), flavonoids by ±13% (5.5 – 6.9), antioxidants by ±5% (53 – 58). Essentially, any conditions provided more or less practicable yield of the biological compounds. The question is, was it worthwhile performing optimization at all? Especially by using advanced techniques such as response surface methodology, neural networks and genetic algorithm.

It looks like the actual purpose of this study was different: not to optimize (the optimization that was done has no big practical value), but to show the feasibility of applying certain optimization techniques to a certain task. The authors have done this professionally. But the goal setting part of the manuscript must be reformulated.

2.    In the Abstract and Table 2 the authors claim that they changed three parameters: the amplitude of ultrasound  (30-60%), treatment time  (5-15  min), and ethanol  concentration  (40-60%). But in section 2.4 they say that they changed “ultrasonic amplitude (20%, 30%, 40%, 50%, and 60%), ethanol concentration (20%, 40%, 50%, 60%, 80%, and 100%),  and solvent to solid ratio (10:1, 20:1, 30:1,  40:1,  and 50:1)”. Why treatment time was excluded from one-factor-at-a-time experiments, but solvent-to-solid ratio was added? (but never mentioned afterwards!)

 3.    Abbreviations in the title is bad manners.

Author Response

Response to Reviewer 1 Comments

Point 1: The authors changed the three parameters in rather wide ranges: the amplitude of ultrasound (30-60%), treatment time (5-15 min), and ethanol concentration  (40-60%). But according to Table 2, the results changed very little in response to these changes of conditions: phenols by ±3% (9.4 – 9.9 mg\g), flavonoids by ±13% (5.5 – 6.9), antioxidants by ±5% (53 – 58). Essentially, any conditions provided more or less practicable yield of the biological compounds. The question is, was it worthwhile performing optimization at all? Especially by using advanced techniques such as response surface methodology, neural networks and genetic algorithm.

It looks like the actual purpose of this study was different: not to optimize (the optimization that was done has no big practical value), but to show the feasibility of applying certain optimization techniques to a certain task. The authors have done this professionally. But the goal setting part of the manuscript must be reformulated.

Response 1: Yes, our primary objective in this study was to compare RSM and ANN-GA in modeling and optimization of complex systems. Based on your comment, we have modified the aim and conclusion of the manuscript to align with our objective. While the optimization results showed limited variation in the yield of biological compounds, our study still provides valuable insights into the feasibility of using these advanced optimization techniques in complex systems. We appreciate your input and hope that the revisions made to the manuscript address your concerns.

Point 2: In the Abstract and Table 2 the authors claim that they changed three parameters: the amplitude of ultrasound (30-60%), treatment time (5-15  min), and ethanol  concentration  (40-60%). But in section 2.4 they say that they changed “ultrasonic amplitude (20%, 30%, 40%, 50%, and 60%), ethanol concentration (20%, 40%, 50%, 60%, 80%, and 100%), and solvent to solid ratio (10:1, 20:1, 30:1, 40:1,  and 50:1)”. Why treatment time was excluded from one-factor-at-a-time experiments, but solvent-to-solid ratio was added? (But never mentioned afterwards!)

Response 2: Thank you for bringing this to our attention. We have updated Section 2.4 to include the treatment time as one of the parameters tested in our OFAT experiment. Additionally, we have included the results and discussion related to OFAT in our manuscript.

Regarding the difference in the parameter ranges, mentioned in the Abstract, Table 2, and Section 2.4. We used a one-factor-at-a-time (OFAT) approach for preliminary screening and determine the range of the independent variables for our experimental design. We had taken a large range of independent variables [ultrasonic amplitude (20%, 30%, 40%, 50%, and 60%), ethanol concentration (20%, 40%, 50%, 60%, 80%, and 100%), treatment time (5, 10, 15 and 20 mins) and solvent to solid ratio (10:1, 20:1, 30:1, 40:1, and 50:1)] for the OFAT based on that it provided useful intial and final range of independent variables [the amplitude of ultrasound (30-60%), treatment time (5-15  min), and ethanol  concentration  (40-60%)] reducing the range of the the variable to find out the optimum condition.

Point 3: Abbreviations in the title is bad manners.

Respone 3: We apologize for that, we have included the full terms instead of abbreviations

Reviewer 2 Report

1- Why used HP LC , for the determination of phenolic and organosulphur, Prefer GC-Ms for the determination all compounds, not only some compounds that you have chosen 

2- Allium sativum please add (L)

3-According to table 1, the levels were ultrasound(30,45 and60), time(4,10 and 15) ethanol (40, 50 and 60), in table 2 in some run time was 2 and 18, ethanol 67, 63

4- in the results Proximate Analysis, please add a table or figure clear that all compounds represent 100% including moisture

5-Lines377, 378 the antioxidant 59.59 do not present in Table 2 please revised and revised conditions

6- Figures 6 and 7 please add the optimization   conditions were used

7- please improve the conclusions, clear the best  optimization condition

Author Response

Response to Reviewer 2 Comments

Point 1: Why used HPLC, for the determination of phenolic and organosulphur, Prefer GC-Ms for the determination all compounds, not only some compounds that you have chosen. 

Response 1: Thank you for bringing up this point of concern. We want to assure you that we have taken it into consideration before going for HPLC approach and conducted experiments accordingly. Mass spectroscopy provide the identified compounds based on generated mass spectra. We used the main available standards of polyphenol and organosulfur compounds of garlic and detected the respective compound qualitatively and quantitatively using HPLC. Allicin is the major bioactive compound formed on crushing of garlic and is decomposed at the injection port of GC and converted to vinyldithiins, thus HPLC detection approach for the organosulfur compounds are preferred. This point has been included in the manuscript in the respective section.

Point 2: Allium sativum please add (L)

Response 2: Thank you for your comment. We appreciate your suggestion to add (L) after Allium sativum to indicate the author of the botanical name. We have made the necessary changes in the manuscript to ensure accuracy and precision.

Point 3: According to table 1, the levels were ultrasound (30,45 and60), time (4,10 and 15) ethanol (40, 50 and 60), in table 2 in some run time was 2 and 18, ethanol 67, 63

Response 3: Yes, you are correct. In some runs, the time was 2 and 18 minutes and ethanol was 33 and 67,  and amplitude as 20 and 70% respectively. These values represent the axial points of the central composite design (CCD), which were selected to estimate the curvature of the response surface. These axial points, also known as alpha points (α), are mentioned in Table 1 along with the levels of the independent variables.

Point 4: in the results Proximate Analysis, please add a table or figure clear that all compounds represent 100% including moisture.

Response 4: Table added.

Point 5: Lines 377, 378 the antioxidant 59.59 do not present in Table 2 please revised and revised conditions

Response 5: Yes, we have revised the conditions and corrected the mentioned value. Thank you for bringing this to our attention.

Point 6: Figures 6 and 7 please add the optimization conditions were used.

Response 6: Thank you for bringing this to our attention. We, have included the optimized conditions that were used for both FE-SEM (figure 6) and FTIR (figure 7) analysis figures

Point 7: please improve the conclusions, clear the best optimization condition

Response 7: Yes, we have updated the conclusions and added the best optimization condition obtained using ANN-GA in the manuscript. Thank you for your feedback.

Round 2

Reviewer 1 Report

The authors have adequately addressed all the queries and the manuscript can be accepted.

Reviewer 2 Report

Accept in present form